# The Contribution of Neutrophils to the Pathogenesis of RSV Bronchiolitis

**DOI:** 10.3390/v12080808

**Published:** 2020-07-27

**Authors:** Ismail Sebina, Simon Phipps

**Affiliations:** Respiratory Immunology Laboratory, QIMR Berghofer Medical Research Institute, Herston 4006, Australia; Simon.Phipps@qimrberghofer.edu.au

**Keywords:** respiratory viruses, RSV, PVM, bronchiolitis, innate immunity, neutrophils, inflammation, cytokines, chemokines, mucosal immunology

## Abstract

Acute viral bronchiolitis causes significant mortality in the developing world, is the number one cause of infant hospitalisation in the developed world, and is associated with the later development of chronic lung diseases such as asthma. A vaccine against respiratory syncytial virus (RSV), the leading cause of viral bronchiolitis in infancy, remains elusive, and hence new therapeutic modalities are needed to limit disease severity. However, much remains unknown about the underlying pathogenic mechanisms. Neutrophilic inflammation is the predominant phenotype observed in infants with both mild and severe disease, however, a clear understanding of the beneficial and deleterious effects of neutrophils is lacking. In this review, we describe the multifaceted roles of neutrophils in host defence and antiviral immunity, consider their contribution to bronchiolitis pathogenesis, and discuss whether new approaches that target neutrophil effector functions will be suitable for treating severe RSV bronchiolitis.

## 1. Introduction

Respiratory syncytial virus (RSV) is the leading viral cause of lower respiratory infections (LRI) among young infants [1,2]. In 2015, an estimated 33 million cases and 120,000 deaths due to RSV infection were reported worldwide [2]. The estimated cost of treating severe cases of RSV-induced bronchiolitis is over $300 million per year [3], and an FDA-approved vaccine against RSV remains elusive [4,5]. A greater understanding of the molecular mechanisms underlying the immunopathogenesis of RSV-bronchiolitis may reveal novel pathways that can be targeted for improving treatment therapies for severe RSV patients. Severe viral bronchiolitis in infancy is a major independent risk factor for the development of chronic asthma in later life [6]. Therefore, ameliorating severe bronchiolitis in infancy may also act as a preventative strategy for later asthma in susceptible individuals. The prevailing view is that severe bronchiolitis results from excessive inflammation and consequent immunopathology, rather than virus-induced cytology. Neutrophils are the dominant inflammatory cell in the airways of paediatric patients with RSV bronchiolitis [7,8,9], accounting for up to 80% of the cellular infiltrate at peak symptomatology [8], and yet the precise contribution of neutrophils to antiviral immunity remains ill-defined.

In the context of anti-bacterial or anti-fungal immunity, neutrophils mediate protection through the production of antimicrobial peptides, respiratory oxygen species (ROS), cytokines and chemokines, and the formation of neutrophil extracellular traps (NETs). New information from omics-based technologies and advances in multi-parameter flow cytometry are shedding new light on neutrophil diversity and function, suggesting a greater complexity to a cell that is often unfavourably viewed as a blunt tool on account of its spectacular prowess at microbial killing and short life-span. Here, we review the multifaceted roles of the neutrophil in host defence, antiviral immunity, and immunopathogenesis in the context of viral LRIs. We find that neutrophils make an important contribution to innate antiviral immunity and help to optimise the induction of an effective adaptive immune response. However, as a consequence of their ability to initiate and amplify the magnitude of an inflammatory reaction, it is clear that a failure to adequately regulate this response can lead to excessive collateral damage and a loss of tissue function, leading to significant morbidity from an RSV infection. We consider a number of novel approaches that might be employed to either harness or suppress the actions of neutrophils to promote antiviral immunity and/or ameliorate the immunopathology associated with severe RSV infection.

## 2. Neutrophils Deploy a Diverse Anti-Microbial Arsenal against Invading Pathogens

An estimated four billion neutrophils are produced in the bone marrow every hour [10,11], and accordingly, neutrophils are by far the most abundant leukocyte in blood, representing approximately 60% of the total white blood cell count. As well as circulating in large numbers, neutrophils enter and surveil tissues in the steady state [11]. Neutrophils are one of the host’s first defensive measures against a sterile or microbial insult. Neutrophils employ three major defensive mechanisms: phagocytosis, degranulation (secreting an array of anti-microbial peptides), and NETosis [12,13]. During phagocytosis, neutrophils utilise a combination of surface-expressed opsonic receptors, intracellular signalling cascades, and cytoskeletal rearrangements to engulf and internalise microbes [14,15]. Neutrophils contain cytoplasmic granules, which store antimicrobial molecules and facilitate cross-talk with other immune cells [16]. These granules (summarised in Table 1) are subdivided into primary or azurophilic granules (containing, e.g., myeloperoxidase, CD63), secondary or specific granules (containing, e.g., lipocalin-2 and lactoferrin), and tertiary or gelatinase granules (containing, e.g., cathelicidins and neutrophil collagenase) [16]. This classification is based on their production during granulopoiesis and expression of distinct protein markers [16]. As a consequence of phagocytosis, the internalised microbe is trapped in ‘bag-like’ structures known as phagosomes, which fuse with neutrophil granules for the safe destruction of microbes [14]. Upon fusion, antimicrobial molecules (summarised in Table 1) are released into the phagosome, whereupon they kill the trapped microbe [16,17].

Neutrophils can also eliminate microbes by releasing extracellular DNA traps composed of decondensed nuclear chromatin, histones, and protein granules, in a process termed NETosis [28,29]. Classical activation of NETosis depends on NADPH oxidase-mediated production of ROS [30], which promotes chromatin decondensation via histone deamination or citrullination. This process is aided by the downstream activation of the protein-arginine deiminase type 4 (PAD4), a nuclear enzyme that citrullinates arginine residues, converting amine groups to ketones [31,32,33]. ROS production can also activate azurophilic granules to release myeloperoxidase (MPO) and neutrophil elastase (NE). If these enzymes translocate to the nucleus, they can disrupt chromatin packaging [15,17], and promote the release of chromatin, which then interacts with other granular and cytosolic proteins to form ‘NETs’ [30]. These NETs are important for immunity against fungi and some bacterial species [22,32,34]. For example, people who are unable to produce NETs develop chronic granulomatous disease and are highly susceptible to invasive aspergillosis [23,35]. Similarly, individuals lacking MPO are susceptible to recurrent fungal infections [23]. In PAD4-deficient mice, neutrophil killing of *Shigella flexneri* or group A *Streptococcus* is diminished [36]. Collectively, these studies demonstrate that NETosis is an important weapon in the host’s defensive armoury. However, if left unchecked, NET-induced responses may also mediate severe pathology. Excessive NETosis can damage the epithelium in pulmonary aspergillosis [37], damage the endothelium in transfusion-related acute lung diseases [38], and exacerbate rhinovirus infection-induced allergic asthma [39]. Therapeutic approaches to block NETosis are now being assessed for the treatment of infectious diseases (including bacterial sepsis), autoimmune diseases (including systemic lupus erythematosus [40], rheumatoid arthritis [41]) and chronic lung disorders (such as cystic fibrosis [42]).

## 3. Neutrophils Influence Innate and Adaptive Immunity

Neutrophils are a vital component of the innate immune system, with key roles in pathogen recognition, the killing of invading pathogens, the presentation of antigens to T-cells, the recruitment of other inflammatory cells, and the production of cytokines [18,43]. For sensing invading pathogens, neutrophils employ a vast array of pattern recognition receptors (PRRs), including Toll-like receptors (TLRs), C-type lectin receptors (e.g., Dectin-1) and cytoplasmic sensors of ribonucleic acids (RIG-I and MDA5) [18,44,45]. Neutrophils also express nucleotide-binding oligomerisation domain (NOD)-like receptors, important for inflammasome formation [46]. The sensing of pathogens through these PRRs activates the effector functions of neutrophils, including ROS production, NET formation, and degranulation (as highlighted in Section 2). Activated neutrophils may also influence the quality of innate and adaptive immune responses by affecting the trafficking and function of other innate cells, such as dendritic cells (DCs) [47,48,49,50,51,52,53]. For example, the neutrophil-derived chemokine CCL3 supports the rapid recruitment of DCs to the inoculation site during *Leishmania major* infection [54]. DC function can also be affected: in the context of an infection with *Toxoplasma gondii,* neutrophil-depletion attenuated interleukin (IL)-12 and TNF production by splenic DCs [55]. Neutrophils also affect the recruitment of other cells too, which they accomplish via the secretion of pro-inflammatory mediators such as danger-associated molecular patterns (DAMPs; e.g., high mobility group box 1 (HMGB1), double stranded DNA, and S100 complex proteins), pro-inflammatory cytokines (e.g., IL-1α, IL-1β, IL-6, IL-17, and TNF) and chemokines (e.g., CXCL1, CXCL2, CXCL8, CCL2, and CCL3) [12,56,57].

Neutrophils can also initiate adaptive immune responses by directly presenting antigens themselves, or by acting as accessory cells, to support T-cell responses. For instance, human neutrophils upregulate MHC-II and co-stimulatory molecule (CD40 and CD80) expression and can present antigens to CD4^+^ T-cells following phagocytosis [58]. Neutrophil acquisition of these antigen-presenting properties (MHC-II, CD40, and CD80 expression) is associated with increased activation and proliferation of CD4^+^ T-cells in response to tetanus toxoid. Neutrophils can also direct T-cell recruitment to the site of infection. For example, in response to influenza virus infection, lung-infiltrating neutrophils were found to deposit a long-lasting chemoattracting trail (expressing the chemokine CXCL12) in the lung to guide antigen-specific CD8^+^ T-cells into specific niches [59]. In the absence of neutrophils, influenza-specific CD8^+^ T-cells were lower in the lung, leading to increased viral load and delayed viral clearance.

Neutrophils can also influence CD4^+^ T cell helper (Th) responses, in particular, Th17 immune responses [50,51,60]. In a mouse model of allergic asthma, neutrophil cytoplasts (enucleated cell bodies) augmented DC-mediated Th17 responses in the lymph nodes, which subsequently increased asthma-like pathology in the lung [51]. Collectively, these studies demonstrate that neutrophils are important participants in innate immunity and contribute to effective adaptive immune responses.

## 4. The Pathophysiology of RSV Bronchiolitis

The vast majority of human respiratory viruses, including RSV, rhinovirus, influenza, coronavirus, adenovirus, and parainfluenza virus, can cause bronchiolitis [61]. However, RSV-induced bronchiolitis is the leading cause of hospitalisation and death among infants within the first two years of life [1,62]. RSV is highly contagious and persists outside of the host for almost six hours [63]. This prolonged survival facilitates its spread to susceptible individuals, mainly via inoculation of open mucous membranes lining the eyes and buccal cavity. Upon inoculation, RSV infects the nasopharyngeal epithelium of the upper respiratory tract, replicates in epithelial cells, and then spreads to the LRT via the bronchiolar epithelium [9,64]. This occurs within 1–3 days post infection, with peak infectivity occurring at 5–7 days post-inoculation [3,65]. During this period, RSV triggers extensive inflammation (characterised by increased neutrophilia and levels of inflammatory cytokines), mucus hypersecretion, and oedema in the airways [1,9,66]. In severe cases, increased mucus production and deposition of cellular debris can occlude the bronchiole lumen, contributing to bronchiolar obstruction, air trapping, and lobar collapse [9]. Patients with severe disease experience dyspnea, wheezing, and cough, the latter often persisting for three or more weeks. Mild cases of bronchiolitis are manageable in outpatient departments. However, severe disease can be life threatening, necessitating urgent mechanical ventilation and admission to intensive care units for some patients. Individuals with severe RSV-induced bronchiolitis during infancy are more likely to develop impaired lung function, recurrent wheezing, and asthma in adulthood [67].

Whether neutrophils play a beneficial or detrimental role during RSV bronchiolitis remains difficult to ascertain clinically due to difficulties in sampling young infants. However, the predominance of neutrophils in the airway wall and alveoli of lung autopsy samples from fatal cases of RSV-related LRI [9,68] or in the airways of paediatric patients (~80% of the total cell infiltrate) with severe RSV-induced bronchiolitis [8] would suggest that they are not innocent bystanders. We postulate that neutrophils promote lung pathophysiology during RSV bronchiolitis through the production of pro-inflammatory cytokines and the releasing of cytotoxic molecules (illustrated in Figure 1). These products, released through distinct cellular processes (e.g., NETosis, degranulation) contribute to mucus hyperproduction, airway epithelial cell (AEC) death and sloughing, and oedema (Figure 1). In RSV-infected mice, the depletion of neutrophils decreases NE, MPO, and MMP9 levels in the airways [69]. Depletion of neutrophils has also been shown to lower TNFα levels and attenuate airway mucin production in the lungs of RSV infected mice [70]. In addition, *in vivo* antagonism of CXCR2 (a chemokine receptor important for neutrophil recruitment into the lung) reduces mucus production in the airways of RSV-infected mice [71]. In vitro, co-culture of RSV-infected AECs with neutrophils increases AEC damage compared with RSV-infected AECs cultured without neutrophils [72]. Collectively, these findings suggest that neutrophils can contribute to pathogenesis; however, a limitation of modelling RSV in mice is that the virus replicates poorly, and hence this is not a particularly tractable system to assess the beneficial properties of neutrophils, such as their antiviral activities.

Infantile severe viral bronchiolitis remains a major risk factor for persistent wheezing and chronic asthma in childhood [73,74]. Therefore, excessive neutrophilic inflammation in infants with severe RSV bronchiolitis may contribute to the onset of type 2 inflammation and cause long-term defects in lung development that predispose susceptible infants to the development of asthma in later childhood. Thus, regulating excessive neutrophilic responses during severe RSV-bronchiolitis might protect against the loss of lung function, curtail the development of allergic sensitisation, and reduce the prevalence of asthma. In light of this, it is important to understand (1) the molecular events governing neutrophil recruitment to the lung, (2) the type of neutrophils recruited to the lung, (3) the molecular signals that control optimal neutrophil activation and function, and (4) whether specific neutrophil subtypes can be targeted therapeutically to improve the management of severe RSV bronchiolitis. We discuss these questions in the sections below.

### 4.1. Neutrophil Inflammatory Mediators and Lung Pathology during RSV-Induced Bronchiolitis

The earliest stages of neutrophil recruitment are initiated by DAMPs (Table 2), including HMGB1, DNA, and histones, which are released by damaged or dying cells [75]. Indeed, HMGB1 is elevated in the nasopharynx of RSV-infected children compared with those infected with other viruses [76]. Animal models of RSV-bronchiolitis have improved our knowledge on the mechanisms of disease development and progression (reviewed extensively in [77,78,79]); however, as pneumoviruses are host-specific, a major limitation with using mice to study human RSV stems from its inability to replicate efficiently in this setting. To better understand the pathogenic processes that underlie viral bronchiolitis, we and others have preferred to inoculate mice with pneumonia virus of mice (PVM), which is orthologous to human RSV [9,80]. Because PVM is a natural pathogen in rodents, a low inoculum dose (e.g., 10 PFU) can be employed, and this allows for greater insights into host–pathogen interactions over time as the virus replicates in the airway epithelium. Importantly, inoculation with a low dose of PVM can induce the more severe pathologies of infant bronchiolitis, such as alveolar epithelial cell apoptosis, bronchial epithelial necrosis, multifocal acute alveolitis, intra-alveolar oedema, haemorrhage, and increased neutrophilia [76,78,81]. These pathological features are similar to those observed in autopsy samples obtained from fatal cases of RSV bronchiolitis [9]. Although there are limitations with the use of PVM (highlighted in [78]), it is proving a useful model to understand the cellular and molecular processes that underlie pathogenesis. Unlike wild-type (WT) control mice, plasmacytoid dendritic cell (pDC)-depleted, Toll-like receptor (TLR)7-deficient, or interferon regulatory factor (IRF)7-deficient neonatal mice develop severe pathology, characterised by increased neutrophilia and lung inflammation in response to acute PVM infection [80,81,82]. In the absence of the TLR7-IRF7 signalling pathway, massive amounts of preformed airway epithelial cell HMGB1 is released into the extracellular environment, increasing the level of neutrophilic inflammation and amplifying tissue pathology, most notably tissue oedema, epithelial sloughing, and cell death [76,80,82,83]. The cell death occurs through programmed necroptosis and further increases the levels of HMGB1, perpetuating the inflammatory response [76]. RSV infection of primary human airway epithelial cells also induces necroptosis-dependent HMGB1 release [76]. Intriguingly, therapeutic inhibition of necroptosis during severe viral bronchiolitis protected mice against the subsequent development of experimental asthma in later-life [76], indicating that severe bronchiolitis is causally associated with subsequent asthma. Examination of the infant bronchiolitis revealed that pharmacological inhibition of necroptosis or immunoneutralisation of HMGB1 decreases the expansion of IL-13-producing type-2 innate lymphoid cells (ILC2s), and prevents alterations to the airway wall such as thickening of the airway smooth muscle (ASM) layer [76,83]. In vitro, ASM cells cultured with IL-33/IL-2-activated ILC2s isolated from the lungs of PVM-infected mice undergo increased cell division, elucidating a cellular and molecular pathway by which excessive inflammation and immunopathology promotes both type 2 inflammation and an aberrant repair response that alters the architecture of the developing lung [83]. Therefore, therapeutic targeting of the HMGB1 signalling axis may act as a novel asthma preventative by dampening ILC2-mediated type-2 inflammation and associated ASM remodelling.

Low circulating pDC numbers in infancy are associated with acute LRIs [84]. Interestingly, the temporal depletion of pDC in neonatal mice predisposes to severe PVM bronchiolitis due to a lack of immunoregulation by regulatory T-cells [80]. The adoptive transfer of splenic regulatory T-cells to infected pDC-depleted neonatal mice decreased HMGB1 and IL-6 levels, epithelial sloughing, and lung neutrophil infiltration. Despite the lack of effect on viral load, the attenuated inflammatory response improved the clinical score. Additionally, the adoptive transfer of regulatory T-cells in early life was sufficient to protect against the later development of experimental asthma [80]. These data suggest that therapies that enhance immunoregulation will both ameliorate the severity of bronchiolitis and break its nexus with the development of asthma.

DAMPs can indirectly promote neutrophilic inflammation by inducing the production of neutrophil-active chemoattractants (Table 2), such as IL-8 [85] and leukotriene B4 (LTB4) [86]. IL-8 levels correlate with neutrophil numbers in the airways and are elevated in nasal washes of RSV-infected children [68]. Moreover, genetic polymorphisms in the IL-8-encoding gene that increase IL-8 production are associated with increased severity of RSV bronchiolitis [87], and pharmacological inhibition of IL-8 in humans reduces neutrophilia and improves clinical outcomes [88]. Levels of the eicosanoid LTB4 are elevated in endotracheal aspirates of infants with RSV bronchiolitis compared with healthy controls [89], and are associated with increased neutrophilic responses [90]. Treatment of RSV-infected mice with the leukotriene inhibitor zileuton reduces cellular inflammation (including neutrophils) in the lung, prevents RSV-induced weight loss and decreases airway pathology compared with untreated control mice [91]. Mice lacking CCL3 or its receptor CCR1 display significantly reduced numbers of airway neutrophils in response to infection with PVM [92,93]. Moreover, the blockade of CCL3/CCR1 in combination with antiviral therapy improves the survival of mice infected with a lethal PVM dose [94].

DAMPs can also promote γδ-T cells [95] and CD4^+^ TH17 cells [96] to produce IL-17A [96,97,98], which is elevated in nasal aspirates from RSV-infected infants in comparison with uninfected controls [99]. IL-17A augments neutrophil recruitment into the lung by stimulating the production of IL-8 and leukotrienes by microvascular endothelial and lung epithelial cells [100,101]. In addition, IL-17 activates p38 MAPK-induced expression of endothelial adhesion markers (E-selectin, VCAM-1, and ICAM-1) on lung endothelial cells, promoting trans-endothelial migration of neutrophils. Lack of IL-17 in RSV-infected mice reduces neutrophil numbers in the lung, limits mucus production, and improves the cytotoxic T-cell (CTL) responses against RSV [99]. However, the mechanism by which IL-17 deficiency enhances CTL responses to RSV remains unknown. Nonetheless, inhibiting IL-17 signalling may prevent neutrophil-induced inflammation and pathology during RSV infection.

### 4.2. Neutrophil Diversity in RSV-Induced Bronchiolitis

Neutrophils have traditionally been viewed as a homogenous cell population. However, it is now appreciated that diversity exists at the molecular, phenotypic, and functional level [11,13,112,113]. This heterogeneity is a consequence of differences in proliferative capacity, maturation status, transcriptional and epigenetic properties, and environmental cues in tissues [11]. In one study that employed Fucci-474 reporter mice to identify cells at different stages of the cell cycle, subsequent mass cytometry and transcriptomic analysis revealed the existence of three distinct neutrophil subsets in the bone marrow [114]. These subsets consisted of a highly proliferative precursor that differentiated into two non-proliferating subsets, one immature and one mature. The precursor neutrophils were Ly6G- and CXCR2-negative, but expressed high levels of cKit and CXCR4. The immature subset displayed low levels of surface Ly6G and CXCR2 and was positive for CXCR4. The mature subset displayed higher levels of Ly6G, CXCR2, and CD101 expression and lacked surface CXCR4. Immature and mature neutrophil subsets were subsequently identified in human blood, with immature neutrophils associating with increased tumour burden [114], suggesting that neutrophil heterogeneity influences disease outcomes.

Importantly, different neutrophil subsets have been identified in circulation and in the lung during acute RSV-bronchiolitis [115,116]. Flow cytometric analysis revealed four distinct neutrophil subsets according to their differential expression of CD16 and CD62L [115]. These included a mature subset (defined as CD16^hi^CD62L^hi^), an immature one (CD16^lo^CD62L^hi^), a suppressive subset (CD16^hi^CD62L^lo^), and a progenitor subset (CD16^lo^CD62L^lo^). The relative frequency of these subsets changed over the course of infection, suggesting that they may play different roles, although functional differences between these subsets remain to be shown. In a separate study in RSV-infected patients, Geerdink and colleagues identified that neutrophils in BALF, compared to those in blood, displayed heightened expression of the inhibitory immune receptor leukocyte-associated immunoglobulin-like receptor-1(LAIR-1) [117]. Flow cytometry analysis also revealed increased Sirp-α, Siglec-9, CD11b, and reduced CD62L and CD31 expression on activated LAIR-1-expressing neutrophils. Agonistic ligation of LAIR-1 was shown to limit NET formation by BALF neutrophils in vitro. Of note, RSV-infected LAIR-1-deficient mice present with greater neutrophilic inflammation upon inoculation with RSV [118], supporting the notion that agonism of LAIR-1 is a logical strategy to ameliorate the severity of viral bronchiolitis. In a separate study performed in infants with an RSV infection, the neutrophils in the nasopharyngeal aspirates were shown to exhibit lower CD62L and CD31 (PECAM-1) and higher CD54 (ICAM-I), CD11b, and CD18 expression compared to those in peripheral blood [116]. CD18 and ICAM-1 expression on neutrophils are also increased in RSV-infected infants compared with uninfected control infants [119]. In a more recent study, the addition of physiological concentrations of neutrophils to human RSV-infected nasal cultures *in vitro* increased epithelial cell damage, characterised by lower ciliary activity, cilium loss, less tight junction expression, and greater detachment of epithelial cells, compared to cultures with RSV-infected epithelial cells alone [120]. High-throughput analysis of lung-infiltrating neutrophils using the latest transcriptomic and high-dimensional flow cytometry is now needed to better identify different neutrophil subsets and associate these with beneficial and deleterious outcomes. Such knowledge, together with an understanding of the molecular signals controlling neutrophil recruitment and function, is likely to reveal novel prognostic biomarkers and targets for therapeutic intervention.

### 4.3. Are Neutrophils Beneficial in Host Immunity against RSV Infection?

Neutropenic individuals are highly susceptible to bacterial and fungal infections. The evidence with regard to viral infections is less clear [18], suggesting that neutrophils are not critical participants. However, neutrophils do appear to be required for optimal host immunity against an RSV infection. To accomplish this, neutrophils detect virus-associated molecular patterns through pattern recognition receptors (such as TLRs), produce an array of antimicrobial products, and assist the adaptive immune responses. For example, TLR4 expression on neutrophils is important for generating optimal immunity against RSV infection [121,122]. The expression of TLR4 is lower on blood and airway neutrophils of infants with severe RSV infection compared with uninfected controls [122]. In humans, two TLR4 gene mutations are associated with an increased risk of developing severe RSV bronchiolitis [123]. Of note, the beneficial or detrimental roles of TLR-signalling in neutrophils during respiratory viral infections may be context-dependent. For instance, TLR4-signalling may be detrimental in RSV infection but protective against influenza. Therapeutic TLR4 antagonism lowers TNF-α, IL-1α, Ptgs2, and Cxcl1 gene expression and improves survival of mice infected with a lethal dose of influenza A [124].

Some neutrophil anti-microbial products may be exploited for boosting anti-viral immunity during RSV infection. For instance, inflammatory neutrophils produce cathelicidins; cationic proteins important for host defence. Intriguingly, in infants hospitalised with viral bronchiolitis, those with the lowest levels of LL-37 were more likely to be infected with RSV, suggesting that LL-37 is beneficial [125]. Human neutrophils stimulated *in vitro* with RSV virions secrete LL-37, inhibiting the formation of new viral particles and reducing the apoptosis of infected epithelial cells [126,127]. Neutrophil-derived LL-37 may support the production of interferons, which are required for viral control. In mice, treatment with an exogenous LL-37 peptide protected against body weight loss, increased interferon-lambda production, and lowered RSV load [128].

Neutrophils may also influence the quality of adaptive immune responses that develop during RSV infection. For example, neutrophils may aid in the rapid recruitment of virus-specific CD8^+^ T-cells. In infants with RSV infection, a systemic influx of bone marrow-derived neutrophil precursors in blood precedes robust CD8^+^ T-cell activation and a reduction in severe disease symptoms [129]. However, the precise mechanisms through which neutrophils influence CD8^+^ T-cells remain unclear. Collectively, these studies suggest that neutrophils can act beneficially in mediating immune protection against RSV infection. However, perturbation of optimal neutrophil function during infection may amplify their effector responses and thus contribute to the pathology that occurs in severe RSV patients.

### 4.4. Are Neutrophils Deleterious during Severe RSV Bronchiolitis?

Excessive neutrophilia is associated with increased tissue damage during severe RSV bronchiolitis [130]. In part, this is attributed to molecular dysregulation of neutrophil-effector pathways, including ROS production, NETosis, degranulation, and over-secretion of proteolytic enzymes [101,111,131,132,133,134], (highlighted in Figure 1). Excessive ROS production activates indiscriminate oxidative stress in the lung, contributing to lung damage [108,109,110]. Indeed, markers of oxidative stress are elevated in the lungs and blood of patients with severe RSV [108]. *In vivo* treatment of mice with antioxidants significantly reduces RSV-induced inflammation and pathology [108].

NE levels are heightened in nasal aspirates and serum from paediatric patients with acute RSV infection compared with uninfected controls [68,106], and NETs are readily detected in BALF samples obtained from patients with an RSV infection [111]. In another study that sampled patients with RSV bronchiolitis, blood neutrophils showed little NET formation, whereas neutrophils from the airways underwent NETosis [117]. Intriguingly, in an ex vivo model, this process was diminished, with an agonistic antibody directed against LAIR-1, which was shown to be elevated on airway neutrophils but not on circulating neutrophils [117]. *In vitro* stimulation of human neutrophils with RSV virions induces ROS-dependent NETosis via PAD-4 citrullination and downstream activation of the PI3K/AKT, ERK, and p38 MAPK pathways [131]. The deposition of NET products in the culture was shown to trap RSV virions in DNA lattices coated with NE and MPO, implicating a beneficial antiviral effect of NETs [131]. However, in calves with a severe bovine RSV infection, the widespread airway obstruction is caused by luminal plugs composed of mucins, cellular debris, and NETs [111]. As DNA-rich mucus is associated with airway obstruction [111,121], excessive NET release likely contributes to pathogenesis, although this has yet to be demonstrated clinically or in an experimental animal model of bronchiolitis. Findings from other experimental systems such as allergic asthma have demonstrated that NETs can exacerbate immunopathology [39]. Virus-associated exacerbations of asthma are attenuated following inhibition of NETosis, either through blockade of NE or degradation of NETs with DNase [39]. Mechanistically, NETosis inhibition reduced type 2-mediated allergic inflammation and lowered airway hyper-reactivity [39]. Clearly, as with most inflammatory processes, NETosis is a double-edged sword. A greater understanding of NETosis in the context of acute LRIs is needed; however, on balance it appears that limiting NETosis in severe RSV bronchiolitis would be advantageous.

Neutrophil-derived toxic granules and proteolytic enzymes also contribute to the pathogenesis of RSV bronchiolitis. RSV increases the expression of matrix metalloproteinase-9 (MMP-9), which augments the influx of neutrophils to the site of infection [107]. Elevated levels of MMP-9 are detected in the respiratory secretions of RSV-infected paediatric patients and are associated with increased risk of requiring mechanical ventilation [133,134,135]. Hence, MMP-9 activity is considered a useful predictor of disease severity in children with RSV-induced respiratory failure [133]. Genetic MMP-9 deficiency in mice decreases viral burden and lung inflammation [134], although it is still unclear whether neutrophil-derived MMP-9 plays a dominant role in the MMP-9-induced pathology during RSV infection. Specific ablation of MMP-9 in neutrophils using Cre-recombinase-LoxP systems would help address this question. In summary, severe neutrophilia and excessive production of powerful antimicrobial mediators appears to cause significant collateral damage, resulting in airway obstruction and the clinical symptoms that characterise severe RSV bronchiolitis.

### 4.5. Therapeutic Regulation of Neutrophil-Induced Pathology in RSV Bronchiolitis

Pharmacological tools to inhibit the excessive neutrophilic response in the lung are likely to ameliorate the severity of RSV bronchiolitis. To achieve this goal, therapeutic strategies should seek to modify the expression of cytokines and/or chemokines, which regulate neutrophil trafficking/recruitment into the lung, and neutrophil activation and function in response to infection (as proposed in Figure 2). Macrolide antibiotics, in particular azithromycin, which is efficacious against IL-8 and neutrophil-induced inflammation, decrease the severity of symptoms during viral bronchiolitis [88,136,137]. A randomised trial in infants hospitalised with RSV bronchiolitis demonstrated that, compared with the placebo-control group, a 14-day treatment with azithromycin significantly decreased wheezing episodes, time to recovery, and overall respiratory morbidity over the subsequent year in comparison [88]. Notably, this was associated with lower IL-8 levels in nasal lavage fluid, suggesting that neutrophil numbers were attenuated, although this was not specifically assessed [88]. Preclinical studies in mice have also demonstrated that azithromycin decreases virus-induced neutrophil accumulation in the lung by abrogating the expression of neutrophil inflammatory mediators such as CXCL1 [136]. In a follow-up study to the clinical trial, azithromycin treatment modified the composition of the upper airway microbiome, reducing the abundance of *Moraxella* [138]. This phenotype was associated with lower odds of developing recurrent wheezing episodes over the next 12 months [138]. Ongoing phase II clinical trials are evaluating whether the addition of azithromycin to routine bronchiolitis care in hospitalised infants reduces the frequency of recurrent wheeze episodes during preschool years (ClinicalTrials.gov identifier: NCT01486758).

Targeting chemokine receptors, for instance CXCR2, which is important for neutrophil migration, may also be a therapeutic option (Figure 2). Danirixin, a selective and reversible antagonist of CXCR2, decreases the activation and transmigration of neutrophils to the lungs of rats treated with aerosolised lipopolysaccharide [139]. A clinical trial to evaluate the efficacy of Danirixin in RSV-infected infants <2 years of age (ClinicalTrials.gov identifier: NCT02201303) is ongoing. Pharmacological inhibition of MMP activity has also been employed as a therapeutic strategy for treating chronic lung diseases [25,140,141]. For example, marimastat, a broad-spectrum MMP inhibitor, reduces bronchial hyperresponsiveness to inhaled allergens in atopic asthmatic individuals [140]. Similarly, mice treated with another broad-spectrum MMP inhibitor, R-94138, also exhibited reduced airway inflammation, characterised by reduced eosinophils and lymphocytes recruited to the lung airways during experimental allergic asthma [141]. As MMP-9 levels are significantly elevated in RSV-infected children [133,134,135], selective inhibition of MMP-9 activity may be of significant benefit for the treatment of RSV bronchiolitis (Figure 2).

Inhibition of NETosis may also be a therapeutic option. Strategies to achieve this might involve the targeting of neutrophil granules (including elastases or MPO), neutrophil-activating pattern recognition receptors (e.g., TLRs) or enzymes (NADPH-oxidase and PAD4). The NE inhibitor AZD9668 improves lung function in patients with bronchiectasis [142]; however, it is yet to be tested for efficacy in infants with a severe RSV infection. Similarly, selective inhibition of MPO with PF-1355, which attenuates tissue injury in experimental immune complex-mediated alveolitis [143], may offer a therapeutic benefit against severe RSV bronchiolitis (Figure 2). The detection of pathogen-associated molecular patterns by pattern recognition receptors on neutrophils can induce NETosis. Therefore, pharmacological disruption of these interactions may attenuate NETosis. For instance, TAK-242, a small molecule TLR4 inhibitor [144] that reduces inflammation, may also be used to regulate NETosis. Considering that the formation of NETs is largely dependent on the activity of NADPH oxidase or PAD4, the inhibition of either enzyme is likely to decrease the magnitude of neutrophil-associated tissue damage. Genetic deletion of PAD4 or PAD4 inhibition by Cl-amidine or BB-Cl-amidine improves clinical outcomes in several different mouse models of inflammatory disease [40,145,146], although whether PAD4 inhibitors are beneficial against a severe RSV infection remains unknown.

The modulation of neutrophil numbers or function is evidently a promising target to ameliorate the severity of RSV bronchiolitis. However, several feasibility issues remain, particularly in relation to safety and specificity. Selective targeting of neutrophils is challenging due to their close relationship to other myeloid lineages, such as monocytes and macrophages. This may alter tissue homeostasis and immune defence against other intracellular pathogens such as *Mycobacterium tuberculosis* or *Listeria monocytogenes*. Moreover, since neutrophils are important for robust antimicrobial host defence, their therapeutic inhibition may predispose the host to other infectious agents or effect colonisation by commensals, which can become pathobionts if left unchecked. To overcome this, one strategy would be to target specific neutrophil subsets, and thus ablate the pathogenic population without compromising the beneficial effect of neutrophils in mediating antimicrobial host defence. New insights are needed to identify approaches that can support this possibility. Most therapeutic approaches in the pipeline have focused on inhibiting excessive neutrophil function in treating disease pathology. However, an alternative approach is to modify neutrophil function to boost host immunity against pathogenic microbes, particularly in the elderly. For instance, the cholesterol-lowering medication simvastatin has been associated with improved neutrophil function and immunity against bacterial infection in humans [147]. In this study, adding simvastatin to the normal treatment regimen significantly reduced the severity of bacterial pneumonia and sepsis in patients older than 62 years [147]. Mechanistically, simvastatin acted by augmenting the migratory accuracy of neutrophils into tissues, albeit with reduced NETosis and NE release, which promoted optimal non-inflammatory bacterial control. Therefore, pharmacological approaches to rejuvenate neutrophils in the elderly might be beneficial in boosting immunity against RSV.

## 5. Concluding Remarks

There remains a pressing need to develop new treatments to reduce the significant morbidity and mortality associated with severe RSV bronchiolitis. Neutrophils contribute to host immunity against RSV; however, they are strongly associated with the development of severe disease and are thus a logical target for therapeutic intervention. New technological approaches, such as single cell RNA sequencing, combined with proteomics, metabolomics, and advances in flow cytometry that allow for multi-parameter analyses, must now be applied to clinical samples from infants with mild and severe disease to gain a greater understanding of the full repertoire of neutrophil heterogeneity and function in the pathogenesis of RSV bronchiolitis. Such information will reveal important insights into neutrophil diversity and identify novel tractable and druggable targets to ameliorate the severity of RSV-bronchiolitis.

## Figures and Tables

**Figure 1 viruses-12-00808-f001:**
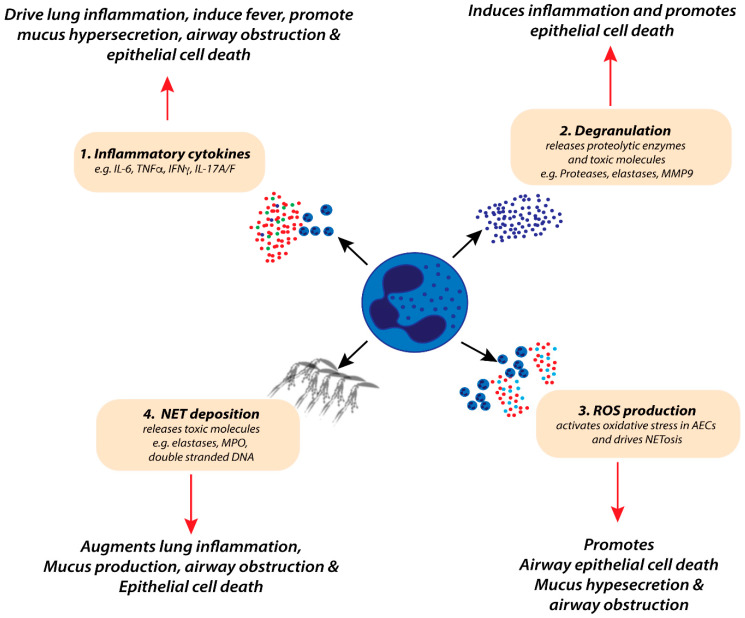
Potential roles of neutrophils in the pathophysiology of severe respiratory syncytial virus (RSV) bronchiolitis. Excessive neutrophil-derived inflammatory cytokine production (1), degranulation (2), respiratory oxygen species (ROS) production (3), and the release of neutrophil extracellular traps (NETosis) (4) are associated with increased lung inflammation, systemic fever, mucus hypersecretion, airway obstruction, and epithelial cell death. Together, these factors contribute to increased lung damage during severe RSV bronchiolitis.

**Figure 2 viruses-12-00808-f002:**
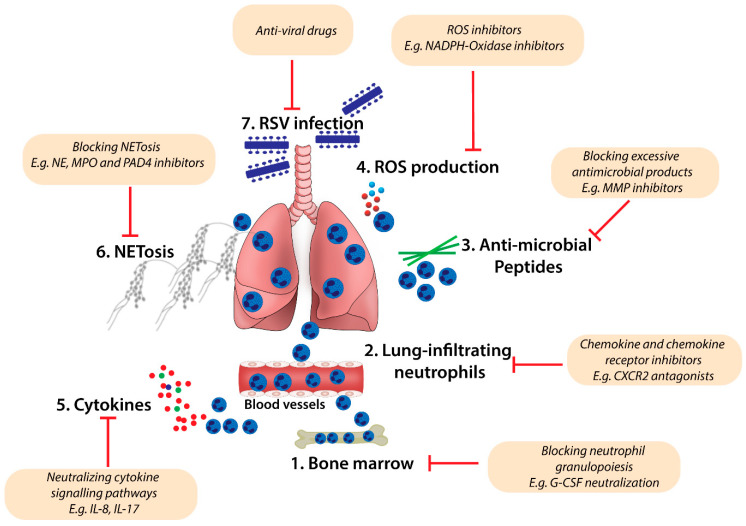
A schematic model for targeting excessive neutrophil homeostasis and function in RSV bronchiolitis. Pharmacological inhibition of excessive neutrophil maturation, recruitment into lung tissues, and effector function may improve the treatment of severe RSV bronchiolitis. Therapeutic inhibition strategies may aim to: (1) block excessive neutrophil granulopoiesis in the bone marrow (e.g., neutralising G-CSF production and function); (2) antagonise chemokine-mediated neutrophil activation and chemotaxis to the lung microenvironment (e.g., using CXCR2 small molecule inhibitors); (3–6) regulate neutrophil function in the lung (e.g., blocking the production of antimicrobial products such as MMPs (3), ROS (4), inflammatory cytokines (5), and deposition of NETs (6)). Therapeutic anti-viral products (7) may also limit RSV-induced recruitment of neutrophils in the lung during infection.

**Table 1 viruses-12-00808-t001:** Neutrophil granules and their anti-microbial properties.

Protein Name	Function
**Azurophil (primary) neutrophil granules**	
Azurocidin	Antibacterial activity (in particular, specific to Gram-bacteria) [16,18]
Neutrophil defensins	Antibacterial, fungicidal, and antiviral activities [19]
Myeloblastin	Serine protease; facilitates transendothelial neutrophil migration [16]
CD63 antigen	Cell surface receptor for TIMP1; activates cellular signalling cascades [20]
Cathepsin G	Serine protease, cleaves complement C3 and has antibacterial activity [21]
Neutrophil elastase (NE)	Modifies the functions of NK cells, monocytes, and granulocytes; inhibits C5a-dependent neutrophil enzyme release and chemotaxis [16,22]
Myeloperoxidase (MPO)	Microbicidal activity against a wide range of organisms [23]
Cap57	Antibacterial activity (Specific to Gram-bacteria) [16]
**Specific (secondary) neutrophil granules**	
Chitinase-3-like protein 1	Important for inflammation [18]
Lipocalin 2	Iron-trafficking; involved in apoptosis, innate immunity, and renal development; limits bacterial proliferation [24]
Lactoferrin	Antimicrobial activity; stimulates TLR4 signalling, binds heparin [16]
**Gelatinase (tertiary) neutrophil granules**	
Matrix metalloproteinase-9 (MMP-9)	Cleaves gelatin types I and V and collagen types IV and V; important roles in leukocyte migration [25]
Ficolin-1	Anti-microbial pattern-recognition receptor
Cathelicidin antimicrobial peptide	Antibacterial activity; cleaved into 2 antimicrobial peptides FALL-39 and LL-37 [26]
Neutrophil collagenase	Degrades fibrillar collagens (type I, II, and III) [27]

**Table 2 viruses-12-00808-t002:** Neutrophil inflammatory mediators released in response to RSV infection. AECs, airway epithelial cells; DAMPs, danger-associated molecular patterns; DCs, dendritic cells.

Mediators	Examples	Potential Pathogenic Effects during Bronchiolitis
Cytokines	1L-1α	Enhances ICAM-1 expression on AECs [102]
	IL-1β	Pro-inflammatory, cell death [103]
	IL-6	Pro-inflammatory, induces fever, induces AEC damage [104,105]
	TNFα	Pro-inflammatory, induces fever, induces AEC damage [69,105]
	IFNγ	Pro-inflammatory, induces fever, induces AEC damage [99]
	IL-17A/F	Pro-inflammatory, augments neutrophil recruitment and activation [100,101]
Chemokines	IL-8	Augments neutrophil chemotaxis to the lung [85]
	CCL3	Recruitment of innate and adaptive leukocytes to the lung, activation of DCs [92,93]
	CXCL12	Recruitment of CD8 T-cells [59]
Neutrophil Granules	MPO	Induces mucus production, oedema and AEC death
	NE	Induces mucus production, oedema and AEC death [68,106]
	MMP-9	Induces lung inflammation [107]
Others	ROS mediators	Induces oxidative stress, AEC death, augment NETosis formation [108,109,110]
	NETosis	Induces mucus hypersecretion and airway obstruction [111]
	DAMPs (e.g., HMGB1)	Induce secretion of pro-inflammatory cytokines, drive ILC2 responses, induce necroptosis and AEC death [76,83]

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
