# Peer review of "The Contribution of Neutrophils to the Pathogenesis of RSV Bronchiolitis"

_viruses, 2020, doi:10.3390/v12080808_

Round 1

Reviewer 1 Report

The manuscript viruses-859943 entitled “The contribution of neutrophils to the pathogenesis of RSV bronchiolitis” presents an overview about the underlying pathogenic mechanisms to acute viral bronchiolitis and a clear understanding of the beneficial and deleterious effects of neutrophils in the inflammation. The authors  describe the multifaceted roles of neutrophils in host defense and antiviral immunity, consider their contribution to bronchiolitis pathogenesis, and discuss whether new approaches that target neutrophil effector function will be suitable for treating severe RSV bronchiolitis.

The review is well structured and organized, however some topics should be clarified.

Why the FDA-approved vaccine against RSV remains elusive? Clarify what is the principle of this vaccine and why it does not allow the intend result.

In the introduction appears the sentence “Severe viral bronchiolitis in infancy is a major independent risk factor for the development of chronic asthma in later life”. The authors should explain in which consists the chronic asthma and clarify the relationship of the RSV infection with the appearance of chronic asthma.

In the end of second topic, the authors present therapeutic approaches to block NETosis, and regarding the MPO or ROS there is any type of therapeutic approach in development?

In the third topic, please clarify the role of neutrophils in the innate immunity.    

Author Response

Response to Reviewer 1

The manuscript viruses-859943 entitled “The contribution of neutrophils to the pathogenesis of RSV bronchiolitis” presents an overview about the underlying pathogenic mechanisms to acute viral bronchiolitis and a clear understanding of the beneficial and deleterious effects of neutrophils in the inflammation. The authors describe the multifaceted roles of neutrophils in host defense and antiviral immunity, consider their contribution to bronchiolitis pathogenesis, and discuss whether new approaches that target neutrophil effector function will be suitable for treating severe RSV bronchiolitis.

The review is well structured and organized, however some topics should be clarified.

Comment 1: Why the FDA-approved vaccine against RSV remains elusive? Clarify what is the principle of this vaccine and why it does not allow the intend result.

Reply: In response to the reviewer’s concern, we have now cited two comprehensive reviews on RSV vaccines (References 4 and 5). We elected not to elaborate on the history of RSV vaccines as it would interrupt the flow at the start of the review (page 1, line 27), and because the review primarily concerns the role of neutrophils in RSV pathogenesis.

Comment 2: In the introduction appears the sentence “Severe viral bronchiolitis in infancy is a major independent risk factor for the development of chronic asthma in later life”. The authors should explain in which consists the chronic asthma and clarify the relationship of the RSV infection with the appearance of chronic asthma.

Reply: We apologise, but we are unclear what the reviewer means by ‘The authors should explain in which consists the chronic asthma’. However, we have provided some extra comments in the review in relation to the association between severe viral bronchiolitis in infancy and the later development of asthma. Please see text lines 174-180.

Comment 3: In the end of second topic, the authors present therapeutic approaches to block NETosis, and regarding the MPO or ROS there is any type of therapeutic approach in development?

Reply: Compounds that inhibit NETosis have been tested for efficacy in the setting of various autoimmune diseases and cancer. With regard to respiratory medicine, the NE inhibitor (AZD9668) has been shown to improve lung function in patients with bronchiectasis and an MPO inhibitor (PF-1355) has been shown to ameliorate experimental alveolitis. This information is provided at lines 423-427. To our knowledge, similar preclinical/clinical studies have yet to be trialled in the setting of severe RSV bronchiolitis.

Comment 4: In the third topic, please clarify the role of neutrophils in the innate immunity.    

Reply: As suggested by the reviewer, we have now revised section 3 to include greater detail on the role of neutrophils in innate immunity. In particular, we discuss their role in pathogen recognition, their innate-killing mechanisms, cytokine production and their ability to present antigens to T cells. We also highlight their roles in recruiting and activating antigen presenting cells (such as dendritic cells). This information is provided in text lines 94-102.

Reviewer 2 Report

Sebina and Phipps tried to describe different aspects of neutrophil contributions to RSV-induced bronchiolitis. Overall, this review manuscript is well written. However, this reviewer has some concerns.

Major concerns:

  1. This review manuscript does not appear to provide clear evidence (by accumulating previously published paper) on how the neutrophils contribute to RSV-induced severe disease pathophysiology (e.g. bronchiolitis), which is necessary as per title. However, it is apparent that the neutrophils were identified during acute RSV bronchiolitis.
  2. Figure 1 is confusing. For readers’ convenience, the authors may provide where and how neutrophil involve in RSV infection and pathophysiology.
  3. As cytokines and chemokines are the key regulatory factors, a figure is required to show clearly these immunomodulators’ (induced by either infected-cell, neutrophils, or the other cell types) pathways for orchestrating bronchiolitis.
  4. NETosis is a novel idea, which is great but it needs more clarification with evidence on its role in bronchiolitis.
  5. The authors described neutrophil contributions to bronchiolitis (e.g. section 4.1) based on data from pneumonia virus of mice (PVM) model, which is RSV-related mouse virus. For the readers’ convenience, a clear distinction, relevant information, and rationale for describing PVM are required. Additionally, a table would be helpful comparing neutrophil contributions to bronchiolitis in other RSV-related viruses, e.g. human metapneumovirus (HMPV), human parainfluenza virus type 3 (HPIV3).
  6. The age-associated factor is important in RSV-induced bronchiolitis, the authors may explain how the age-related factors contribute to neutrophilia and its role RSV-induced bronchiolitis. For example, whether neutrophil plays any role in RSV-induced long-term consequential pathogenesis, e.g. pediatric asthma.

Minor concerns:

  1. For readers’ convenience, the term neutrophil inflammation needs to be explained.
  2. The authors described a different subset of neutrophils. A figure would helpful describing each subset’s role in bronchiolitis, pathogenesis, or viral clearance.
  3. A connection needs to be made about how the nasal respiratory response with respect to neutrophil corresponds to bronchiolitis.

Author Response

Response to Reviewer 2

Sebina and Phipps tried to describe different aspects of neutrophil contributions to RSV-induced bronchiolitis. Overall, this review manuscript is well written. However, this reviewer has some concerns.

Major concerns:

Major concern 1: This review manuscript does not appear to provide clear evidence (by accumulating previously published paper) on how the neutrophils contribute to RSV-induced severe disease pathophysiology (e.g. bronchiolitis), which is necessary as per title. However, it is apparent that the neutrophils were identified during acute RSV bronchiolitis.

Reply: Clinical evidence implicating a deleterious role for neutrophils in severe RSV bronchiolitis is primarily associative (discussed in sections 4.3 and 4.4). In response to the reviewer’s concern, we have revised the manuscript to highlight possible mechanisms by which excessive neutrophil-responses (via ROS production, NET formation, cytokine production and secretion of proteolytic enzymes etc.) might contribute to increased mucus production, necrosis of airway epithelial cells, airway obstruction and airway smooth muscle remodelling, thus driving part of the pathology that occurs during RSV bronchiolitis. This information is provided in text lines 160-173. These pathways are also depicted on a newly added figure (please see Figure 1) to the manuscript.

Major concern 2: Figure 1 is confusing. For readers’ convenience, the authors may provide where and how neutrophil involve in RSV infection and pathophysiology.

Reply: As per the reviewer’s suggestion, we have now generated a new figure (Figure 1), highlighting the pathophysiological roles of neutrophils during RSV bronchiolitis. To provide the reader with greater clarity, we have annotated the figure to include various neutrophil pathways activated during infection and their potential injurious roles. Figure 2 (previously, Figure 1), now depicts potential therapeutic approaches to regulating neutrophil-induced lung pathology during RSV bronchiolitis.

Major concern 3: As cytokines and chemokines are the key regulatory factors, a figure is required to show clearly these immunomodulators’ (induced by either infected-cell, neutrophils, or the other cell types) pathways for orchestrating bronchiolitis.

Reply: As suggested by the reviewer, we have now added a new table to the manuscript (please see table 2), summarising neutrophil-derived cytokines, chemokines and other mediators secreted during bronchiolitis.

Major concern 4: NETosis is a novel idea, which is great but it needs more clarification with evidence on its role in bronchiolitis.

Reply: The role of NETosis in the context of viral bronchiolitis, is poorly defined. Nonetheless we discussed the latest literature available and revised the text for greater clarity as suggested by the reviewer. This information is provided in text lines 343-364.

Major concern 5: The authors described neutrophil contributions to bronchiolitis (e.g. section 4.1) based on data from pneumonia virus of mice (PVM) model, which is RSV-related mouse virus. For the readers’ convenience, a clear distinction, relevant information, and rationale for describing PVM are required. Additionally, a table would be helpful comparing neutrophil contributions to bronchiolitis in other RSV-related viruses, e.g. human metapneumovirus (HMPV), human parainfluenza virus type 3 (HPIV3).

Reply: As suggested by the reviewer, we have now provided the rationale for the use of PVM as a model system in mice, thereby prefacing the following section detailing some of the findings from this model. Please see modified text in 189-202.

While dysregulated neutrophilic inflammation is common to all three virus infections (PVM, HMPV and HPIV3), there is insufficient evidence to compare and contrast the function of neutrophils between each of these settings. Therefore, we have elected not to include such a table.

Major concern 6: The age-associated factor is important in RSV-induced bronchiolitis, the authors may explain how the age-related factors contribute to neutrophilia and its role RSV-induced bronchiolitis. For example, whether neutrophil plays any role in RSV-induced long-term consequential pathogenesis, e.g. pediatric asthma.

Reply: As suggested by the reviewer, we have now revised the text, highlighting a potential mechanism through which neutrophilic inflammation that occurs during early life severe RSV bronchiolitis may predispose susceptible infants to asthma in childhood. This information is provided in text lines 174-180.

Minor concerns:

Minor concern 1: For readers’ convenience, the term neutrophil inflammation needs to be explained.

Reply: Neutrophilic inflammation describes an inflammatory response where neutrophils are prevalent. We do not believe this requires an explanation.

Minor concern 2: The authors described a different subset of neutrophils. A figure would helpful describing each subset’s role in bronchiolitis, pathogenesis, or viral clearance.

Reply: In section 4.2, we discuss the presence of different subsets of neutrophils, identified in blood and nasal aspirates of RSV patients. As stated in the manuscript, the specific role of these different subsets in bronchiolitis remain unknown, and therefore, it is not possible to include a figure describing the roles of particular subsets. However, in the course of revising the manuscript, we identified some relevant literature and have now revised the section to make it more comprehensive.

Minor concern 3: A connection needs to be made about how the nasal respiratory response with respect to neutrophil corresponds to bronchiolitis.

Reply: In section 4, text lines 135-140, we have discussed the clinical manifestation of RSV, with RSV binding to and infecting epithelial cells in the upper airways, before spreading to the lower respiratory tract. While in the upper airways, it replicates rapidly, causing epithelial cell necrosis and ciliary destruction. This triggers a cascade of pro-inflammatory responses, characterised by an influx of airway infiltrating neutrophils and increased production of inflammatory cytokines. In response to the reviewer’s concern, we have also modified the text in lines 139-140 to highlight a link between neutrophils in the upper airways and bronchiolitis.

Reviewer 3 Report

This is a well-written comprehensive review that describes the current status on the role of neutrophils in the pathogenesis of RSV bronchiolitis. The authors discuss the challenges and confounding data on neutrophils being both beneficial and deleterious, and how targeting neutrophil effector function could have therapeutic relevance considering vaccines for RSV continues to be a challenge. For most part, references are up to date but whether the literature on neutrophils and RSV infection in animal models is comprehensive is not clear. The schematic makes a good case for the many roles that neutrophils play in response to RSV infection. Authors need to address the following points:

1) ref. 65 refers to a mouse model and mice do not express CXCL8/IL-8. However, in the text, authors cite this reference for ‘pharmacological inhibition of IL-8 in humans reduces neutrophilic inflammation and improves clinical outcomes [64-66]’.

2) It would help if the authors have a separate section for animal models, and also discussion along the lines to what extent mouse models capture human disease and any limitations will be helpful. For instance, authors make a compelling case for IL-8 in the disease etiology in humans. Mice do not express IL-8 but KC/mCXCL1, MIP2/mCXCL2, and LIX/mCXCL5 are the major chemokines that recruit neutrophils. Reference 65 also shows reduced KC levels in antibiotic treated mice. Further, ref.65 uses a Sendai Virus (SeV) and not RSV, and so a discussion on whether SeV captures RSV infection will provide a proper context.

3) In the below sentence, ref. 109 does not discuss CXCL1 levels

Preclinical studies in mice have also demonstrated that azithromycin decreases virus-induced neutrophil accumulation in the lung by abrogating the expression of neutrophil inflammatory mediators such as CXCL1 [65, 109].

4) stating that CXCR2 is an IL-8 receptor is misleading as CXCR1 is the IL-8 receptor. CXCR2 binds all ELR-chemokines including IL-8 with high affinity. Ref. 111 discusses danirixin that binds CXCR2 with much higher affinity compared to CXCR1. Moreover, whereas CXCR1 is highly expressed in human neutrophils, CXCR1 expression is more selective in mice.

Targeting chemokine receptors, for instance CXCR2, the IL-8 receptor, may also be a therapeutic option [111].

5) Under ‘Neutrophils influence innate and adaptive immunity’ section, first para, last sentence – CXCL2 is repeated twice and CXCL8 is missing.

Author Response

Response Reviewer 3

This is a well-written comprehensive review that describes the current status on the role of neutrophils in the pathogenesis of RSV bronchiolitis. The authors discuss the challenges and confounding data on neutrophils being both beneficial and deleterious, and how targeting neutrophil effector function could have therapeutic relevance considering vaccines for RSV continues to be a challenge. For most part, references are up to date but whether the literature on neutrophils and RSV infection in animal models is comprehensive is not clear. The schematic makes a good case for the many roles that neutrophils play in response to RSV infection. Authors need to address the following points:

Comment 1: ref. 65 refers to a mouse model and mice do not express CXCL8/IL-8. However, in the text, authors cite this reference for ‘pharmacological inhibition of IL-8 in humans reduces neutrophilic inflammation and improves clinical outcomes [64-66]’.

Reply: We thank the reviewer for highlighting this error. We have revised the manuscript accordingly. Please see text line 238.

Comment 2: It would help if the authors have a separate section for animal models, and also discussion along the lines to what extent mouse models capture human disease and any limitations will be helpful. For instance, authors make a compelling case for IL-8 in the disease etiology in humans. Mice do not express IL-8 but KC/mCXCL1, MIP2/mCXCL2, and LIX/mCXCL5 are the major chemokines that recruit neutrophils. Reference 65 also shows reduced KC levels in antibiotic treated mice. Further, ref.65 uses a Sendai Virus (SeV) and not RSV, and so a discussion on whether SeV captures RSV infection will provide a proper context.

Reply: In response to the reviewer’s suggestion, we have now cited three comprehensive reviews on animal models of RSV bronchiolitis (References 70-72). We have also provided a brief discussion on the use of pneumonia virus of mice (PVM) as a model system to study the pathophysiology of RSV bronchiolitis. This information is provided in text lines 189-202.

Comment 3: In the below sentence, ref. 109 does not discuss CXCL1 levels

Preclinical studies in mice have also demonstrated that azithromycin decreases virus-induced neutrophil accumulation in the lung by abrogating the expression of neutrophil inflammatory mediators such as CXCL1 [65, 109].

Reply: We thank the reviewer for identifying our error. We have now removed reference 109 from this section.

Comment 4: stating that CXCR2 is an IL-8 receptor is misleading as CXCR1 is the IL-8 receptor. CXCR2 binds all ELR-chemokines including IL-8 with high affinity. Ref. 111 discusses danirixin that binds CXCR2 with much higher affinity compared to CXCR1. Moreover, whereas CXCR1 is highly expressed in human neutrophils, CXCR1 expression is more selective in mice. Targeting chemokine receptors, for instance CXCR2, the IL-8 receptor, may also be a therapeutic option [111].

Reply: We thank the reviewer for identifying our error. We have modified the manuscript to reflect their insights.

Comment 5: Under ‘Neutrophils influence innate and adaptive immunity’ section, first para, last sentence – CXCL2 is repeated twice and CXCL8 is missing.

Reply: We thank the reviewer for highlighting this typographical error. We have modified the text accordingly.

Round 2

Reviewer 2 Report

This revised manuscript has been improved substantially. A few minor suggestions:

  1. Tables 1 and 2: please include references.  
  2. Figure 2: RSV is primarily a filamentous virus. The circular virions can be replaced with filamentous virions.        

Author Response

Reviewer 2 comments

Minor comment 1. Tables 1 and 2: please include references.  

Reply: As suggested by the reviewer, references have now been added to tables 1 and 2.

Minor comment 2. Figure 2: RSV is primarily a filamentous virus. The circular virions can be replaced with filamentous virions. 

Reply: As suggested by the reviewer, Figure 2 graphical circular virions have now been replaced with filamentous ones.